# Therapy-Acquired Clonal Mutations in Thiopurine Drug-Response Genes Drive Majority of Early Relapses in Pediatric B-Cell Precursor Acute Lymphoblastic Leukemia

**DOI:** 10.3390/diagnostics13050884

**Published:** 2023-02-25

**Authors:** Rozy Thakur, Prateek Bhatia, Minu Singh, Sreejesh Sreedharanunni, Pankaj Sharma, Aditya Singh, Amita Trehan

**Affiliations:** 1Pediatric Hematology Oncology Unit, Department of Pediatrics, Advanced Pediatric Centre, Post Graduate Institute of Medical Education and Research, Chandigarh 160012, India; 2Department of Haematology, Post Graduate Institute of Medical Education and Research, Chandigarh 160012, India; 3Department of Cardiovascular Medicine, Stanford University, Stanford, CA 94305, USA

**Keywords:** B-ALL, deep sequencing, therapy-acquired, relapse

## Abstract

Methods: Forty pediatric (0–12 years) B-ALL DNA samples (20 paired Diagnosis-Relapse) and an additional six B-ALL DNA samples (without relapse at 3 years post treatment), as the non-relapse arm, were retrieved from the biobank for advanced genomic analysis. Deep sequencing (1050–5000X; mean 1600X) was performed using a custom NGS panel of 74 genes incorporating unique molecular barcodes. Results: A total 47 major clones (>25% VAF) and 188 minor clones were noted in 40 cases after bioinformatic data filtering. Of the forty-seven major clones, eight (17%) were diagnosis-specific, seventeen (36%) were relapse-specific and 11 (23%) were shared. In the control arm, no pathogenic major clone was noted in any of the six samples. The most common clonal evolution pattern observed was therapy-acquired (TA), with 9/20 (45%), followed by M-M, with 5/20 (25%), m-M, with 4/20 (20%) and unclassified (UNC) 2/20 (10%). The TA clonal pattern was predominant in early relapses 7/12 (58%), with 71% (5/7) having major clonal mutations in the *NT5C2* or *PMS2* gene related to thiopurine-dose response. In addition, 60% (3/5) of these cases were preceded by an initial hit in the epigenetic regulator, *KMT2D*. Mutations in common relapse-enriched genes comprised 33% of the very early relapses, 50% of the early and 40% of the late relapses. Overall, 14/46 (30%) of the samples showed the hypermutation phenotype, of which the majority (50%) had a TA pattern of relapse. Conclusions: Our study highlights the high frequency of early relapses driven by TA clones, demonstrating the need to identify their early rise during chemotherapy by digital PCR.

## 1. Introduction

The relapse of B-cell acute lymphoblastic leukemia (B-ALL), comprising 10–15% of cases, continues to be one of the foremost causes of cancer death in the pediatric population and, despite the application of high-throughput genomics to unravel the biology of the relapse of the disease, gaps exist in evolution trajectories and the nature of relapses due to leukaemic clone heterogeneity and differences between chemotherapy doses in different treatment protocols being used world-wide [1,2,3,4,5,6,7,8,9,10,11,12,13,14,15,16].

Multi-agent chemotherapy is known to induce variable stress on leukaemic clones, leading to clonal evolution and the development of drug-resistant mutations in the genes related to chemotherapy-dose response (*CREBBP, WHSC1*, *NT5C2*, *PRPS1*, *PRPS2*, MSH2, *MSH6*, *PMS2, NR3C1*, *NR3C2, Tp53* and *FPGS*). Such therapy-acquired somatic mutations in chemotherapy-dose-response genes drive >20% of early and late relapses [17,18,19,20,21,22,23]. However, most early relapses occur when the ‘persister clone’ escapes treatment and acquires a second hit, in one of the genes mentioned above, under chemotherapy stress, giving it a proliferative advantage. Such second-hit mutations in these genes are conspicuously absent in the diagnostic sample, even at deep sequencing, and are hence termed treatment-acquired or therapy-acquired mutations.

Hence, the current study was undertaken to address the paucity of data related to the evolution of these therapy-acquired mutations, especially in the context of non-existent data from lower-middle-income countries. We tried to address this by generating bulk sequencing data from paired diagnosis-relapse pediatric B-ALL cases, harnessing the potential of error-read correction through the incorporation of unique molecular barcodes and robust bioinformatic analysis. This study is expected to help us translate better relapse-prediction models and devise residual disease-monitoring strategies by digital PCR during the critical treatment phases of B-ALL.

## 2. Materials & Methods

### 2.1. Patients and Samples

The DNA and RNA of 40 biobanked samples (>95% documented blasts) from 20 pediatric (0–12 years) B-ALL cases (20 paired diagnosis (D)–relapse (R)) who experienced very early (<9 months from diagnosis), early (9–36 months) or late relapse (>36 months) and an additional cohort of 6 non-relapsed B-ALL cases (without relapse >5 years post treatment), as control arm, were retrieved and enrolled for analysis after due approval of institute ethics committee. For excluding germline variants, DNA was obtained from remission sample (post-induction bone marrow) slides. All cases had or were receiving uniform treatment as per ICiCLe protocol with a curative intent [24]. Cases were risk-stratified based on NCI criteria of age and white blood cell (WBC) count and started as per ICiCLe protocol on initial 8 days of prednisolone with a Day 8 absolute blast count (D8 ABC) for categorization as good (GPRs; <1000/µL) or poor prednisolone responders (PPRs; >1000/µL). Induction therapy was further decided based on this day-8 response and cytogenetics data, with standard-risk (SR) cases receiving vincristine and L-asparginase and intermediate (IR) and high-risk (HR) cases 2 and 4 receiving additional cycles of daunorubicin, respectively. Day-35 minimal residual disease (MRD) evaluation was conducted by standard flow cytometry (0.01%) and SR or IR cases, with positive MRD escalated to high-risk consolidation therapy.

Informed written consent was taken from patients’ parents/legal guardians before enrolment in the study. Biobanked DNA and RNA were extracted from peripheral blood mononuclear cells (PBMCs) using Qiagen DNA and RNA blood mini kit (Qiagen, Hilden, Germany), according to manufacturer’s protocol. The quality and quantity of DNA and RNA were checked prior to storage and after retrieval using both Nanoquant and Qubit fluorometer. One µg of total RNA was transcribed into cDNA by superscript cDNA vilo synthesis kit (ThermoFisher Scientific, Waltham, MA, USA) and the quality of cDNA was checked by running a 1.5% agarose gel for PCR product of GAPDH housekeeping gene.

### 2.2. Primary Genetic Subtype Analysis

Primary genetic events in B-ALL were classified by standard triple probe (*ETV6-RUNX1, KMT2A, BCR-ABL1*) FISH panel and DNA ploidy index (FxCycle violet on BD FACS), as per routine testing. Cases negative for above translocations and or ploidy abnormalities were evaluated for *Ph-like* gene expression followed by targeted next generation sequencing (NGS) RNA fusion panel analysis. The *Ph-like* gene expression was assessed by RQ-PCR TaqMan-based array method using a set of pre-validated 15 genes, as reported by Roberts et al. [25] (refer to Appendix A). This is based on scoring expression of genes with a coefficient score >0.5 suggestive of a *Ph-like* expression. The NGS-based targeted RNA fusion panel was designed on Ampliseq studio (hg19) and covered 109 translocations, including isoforms/fusion partners, along with 12 internal control genes with 100% coverage (refer to Appendix A). This targeted panel covers most of the commonly encountered Ph-like fusions involving *JAK-2, ABL1, ABL2, PDGFRA, PDGFRB* and *CSF1R* kinases. Briefly, 50 ng of cDNA was amplified by single primer pool for 27 cycles using the following program: 98 °C for 2 min (1 cycle), 98 °C for 15 s, 60 °C for 4 min (27 cycles), holding at 10 °C followed by amplicon digestion, adaptor ligation and library amplification. Library quantification was performed on Qubit fluorometer, followed by sequencing on Ion Torrent S5 Studio. Fusions were reported as per Ion Reporter software default parameters of >10,000 reads for internal control genes and >25 junctional reads for true translocation call.

### 2.3. Deep Sequencing for Secondary Genetic Abnormalities

Secondary genetic events, such as single-nucleotide variations/insertion-deletions (SNVs/INDELs) and copy-number variations (CNVs), were evaluated using a custom DNA NGS panel of 74 genes based on ArcherDx proprietary chemistry incorporating unique molecular barcodes for error correction (refer Appendix A). The panel comprises all relevant leukemia driver genes in COSMIC/NCI database, as well as a few sourced from research studies. Briefly, 200 ng of high-quality genomic DNA was diluted with 10 mM Tris-HCl pH-8.0 to a total volume of 50 μL. The ArcherDX library reagent kit (ArcherDx Inc., Boulder, CO, USA) was used and libraries were quantified by NEBNext library quant kit for Illumina (New England Biolabs) on Quantstudio3 instrument. Deep sequencing (1050–5000X; mean 600X) was performed on NovaSeq 6000 with at least 10–12 million reads (>2 GB data) per sample. To maintain deep -sequencing depth during analysis and retain singleton reads, singleton correction method of error-read correction, as detailed by Wang et al., was used [26]. Data analysis was performed as per ArcherDx analysis pipeline with somatic filter criteria, as detailed in Appendix A.

### 2.4. Bioinformatic Definitions and Clonal Evolution Patterns and Plots

The variants noted were classified as diagnosis (D)-specific, relapse specific (R) or shared. Clonal mutations were classified as major (>25% VAF) and minor/sub-clonal (<25% VAF), based on assumption that those with VAF >25% have a dominant (>50% clonal cellular frequency (VAF × 2) in clonal composition. The changes in the variant allele frequency (VAF) of somatic SNVs were examined to determine the clonal evolution pattern from diagnosis to relapse. Clonal evolution patterns from diagnosis to relapse were classified into following groups: (i) TA (therapy acquired) for genetically distinct clones at relapse with the major relapse clonal mutation not present in diagnostic sample even at ultra-deep (>3000×) sequencing analysis of the data; (ii) m-M (minor to Major) for clones with minor sub-clones at diagnosis to major sub-clones at relapse (iii) M-M (major to major) for cases in which major clones at diagnosis were the only major clones at relapse (genetically similar clones); and (iv) UNC (unclassifiable), in which no major clones could be identified at diagnosis or relapse. The somatic SNVs were clustered based on their VAFs at diagnosis and relapse and plots were designed using CanvasXpress. The input included the clonal fraction of each tumor cell population, representation of descent in the form of parental relationship and the time points at which the samples were obtained. Hypermutation phenotype in samples (>1.5 mutations per MB) was defined as per definition and criteria proposed by Waanders et al. [27] using CoMutplot software (0.0.3 version).

## 3. Results

The baseline clinical and hematological characteristics of the paired-sample cohort are presented in Table 1. Briefly, the mean age was 6 years, with a M:F ratio 5.6:1. The mean WBC at presentation was 78 × 10^9^/L. The day-35 marrow was M1 (100%); however, seven (35%) had positive MRD (>0.01%) post induction. The final risk stratification post induction was: standard risk 25%, intermediate risk 25% and high risk (50%). The time to relapse ranged from 8–40 months (median 26) with predominant early relapses (60%, 12/20), 25% (5/20) late relapses and 15% (3/20) very early relapses.

The site of relapse was predominantly medullary in 55% (11/20), combined medullary with/out CNS and/or testes in 30% (6/20) and CNS and testicular alone in 10% (2/10) and 5% (1/20), respectively.

### 3.1. Primary Genetic Abnormality Data

A definitive primary genetic abnormality could be defined in 15/20 (75%) cases, both on diagnosis and in paired samples at relapse. These included *ETV6:RUNX1* (4), *BCR:ABL1* (4), *TCF3:PBX1* (1) and hypodiploidy (1) on standard FISH and DNA ploidy analysis and *P2Y8:CRLF2*, *MEF2D:BCL9*, *TAF15:ZNF384*, *EBF1:PDGFRB* and *KMT2A:MLLT1* in one case each on targeted NGS-based RNA fusion analysis. The remaining (5/20; 25%) negative cases after NGS testing were categorized as B-cell other. Overall, *Ph-like* B-ALL genomic lesions were noted in 2/20 (10%) cases. The pie chart in Figure 1 highlights the frequency of primary genetic events in the 20 paired B-ALL cases. In the six in the non-relapse case arm, a primary genetic event was recorded in three (50%) cases, with two *TCF3:PBX1* cases and one *AF4:MLL* case.

### 3.2. Deep Sequencing Data for Secondary Genetic Abnormalities of SNV/INDELs and CNVs in Paired Cases and Non-Relapsed Control Arm Samples

A mean coverage of 1600x was achievable in all the cases and VAFs up to 2.7% can be reported with confidence. After initial bioinformatic data filtering (based on criteria defined in Appendix A), 1552 SNVs/INDELs (D-767 with mean 38; R-785 with mean 39.15) and 37 CNVs (D-15 mean 0.75; R-22 with mean 1.1) were noted in 20 paired B-ALL samples and 234 SNVS/INDELs (mean 39). No CNV was noted in the six non-relapse-arm B-ALL samples (*p*-0.765). After a second data filtration (based on the criteria defined in Appendix A), 235 SNVs/INDELs (D-105 with mean 5; R-130 with mean 6.4) and 43 SNVs/INDELs (mean 7) were noted, respectively (*p*-0.316); Figure 2a). In total, 47/235 (20%) were major mutations and 188/235 (80%) were minor. Of the forty-seven major clones, eight (17%) were diagnosis-specific, seventeen (36%) were relapse-specific and 11 (23%) were shared. Out of 188 minor clones, 62 (33%) were diagnosis-specific, 74 (39%) were relapse-specific and 26 (14%) were shared. In the non-relapse arm, no pathogenic major clone was noted in any of the six samples and all forty-three SNVs/INDELs were minor in nature. A CNV at diagnosis or relapse was noted in 18/20 (90%) cases (shared, 8; gained at relapse, 8; lost, 1; and lost and new gained at relapse, 1), with *CDKN2A/2B* deletion noted in 65% (13/20) and *IKZF1* deletion noted in 25% (5/20). Two or more CNVs were noted in 30% (6/20) of the cases only. The recurrently mutated genes (five or more samples) included *TENM3* (11), *MSH6* (11), *NOTCH1* (10) and *MSH2* (10), with minor clones in all the cases (except one case, with a major *MSH2* clone at diagnosis), followed by *KMT2D* (9), *KRAS* (9), *NRAS* (7), *PMS2* (7) and *EP300* (6), with both major and minor clones (except *EP300*, in which all the clones were minor clones and *PMS2*, which had relapse-enriched major clones; Figure 2b–d).

A total of 48 mutations were noted in common relapse-enriched genes, of which the majority 35/48 (73%) were minor. In total, nine out of twenty (45%) had major clones in one of these twelve genes (*NT5C2*-3, *PMS2*-3, *Tp53*-2 and *PRPS1*-1) driving the relapse, while only two cases had diagnosis-specific major clones. Mutations in the relapse-enriched genes comprised 33% (1/3) of the very early relapses, 50% (6/12) of the early and 40% (2/5) of the late relapses. Multiclonal mutations were noted in nine genes in fifteen cases (75%), with minor clonal *TENM3* or *NOTCH1* being most common (93%; 14/15). On other hand, *KRAS*, *KMT2D*, *Flt3* and *FPGS* had both major and minor clones, either on diagnosis or on relapse (Figure 2d).

### 3.3. Clonal Evolution Pattern

Most of the common clonal evolution patterns observed were therapy-acquired (TA), in 9/20 (45%), followed by M–M 5/20 (25%), m–M 4/20 (20%) and UNC 2/20 (10%). The TA clonal pattern was most predominant in the early relapses, with 7/12 (58%), followed by 2/5 (40%) in late relapses. In addition, two cases with the m–M pattern in very early and early relapse each had a late therapy-acquired clonal gene mutation in the *PRPS1* and *NT5C2* genes, respectively. Overall, TA clonal mutations were noted in six genes, with *NT5C2* and *PMS2* clones in three cases each, followed by *UHRF1* in two cases and *ETV6*, *KRAS* and *PRPS1* in one 1 case each. The very early relapses were primarily m–M (2/3; 67%), originating in minor *RAS* clones. The late relapses had an equal frequency of M–M and TA-type patterns, with *Tp53* clones persisting at diagnosis and relapses in two M–M cases, which is characteristic of *Tp53* clones, since they are known to enter a quasi-dormant state under chemotherapy and later lead to late relapses (Figure 3a–e and Table 1). Two late relapse cases with TA type pattern had *KRAS* and *UHRF1* clones driving relapse.

Overall, 14/46 (30%) samples showed a hypermutation phenotype based on the criteria suggested by Waanders et al. in their study [13] (Table 1). The hypermutation frequency was 13% (6/46) on diagnosis and 17% (8/46) at relapse.

## 4. Discussion

This is one of the first genomic studies in pediatric B-ALL from our sub-continent to address relapse biology and define clonal evolution patterns and therapy-acquired mutations. The study is relatively novel in terms of the error-read correction technology used and the integrated SNV/INDELs and CNV detection performed using a targeted deep sequencing NGS approach. The study cohort of 20 paired diagnosis–relapse B-ALL cases had a heterogenous distribution of primary genetic abnormalities, which was defined in 75% (15/20) of the cases. The *BCR:ABL* and *ETV6:RUNX1* comprised the majority of the cases (53%; 8/15), but if we combine these primary abnormalities into standard prognostic groups, nine (60%) were at lower risk and all of them had either very early or early relapse, while six (40%) were at high-to-intermediate risk and had early-to-late relapses.

No significant difference in the mutational burden of secondary abnormalities of SNVs/INDELs and/or CNVs in the diagnostic and relapse samples was observed (Figure 1a), which was similar to the non-statistical association highlighted by Ma et al. [28] in their study. Furthermore, consistent with the study by Spinella et al. [25], 95% and 75% of our cases had ≥1 mutational event in the epigenetic and *RAS* pathway genes, respectively. Moreover, the *RAS* gene mutants showed heterogenous clonal evolution consistent with their dual nature as sensitive to vincristine and resistant to methotrexate, thereby driving the balance in either direction in individual cases [23,28,29]. In a study by Kuiper et al. [30], around 80% of the RAS pathway gene mutations at relapse could be traced back to a paired diagnostic sample and 40% of them were present as minor sub-clones. In our study, in 75% (6/8) ff the cases, RAS pathway mutations could be traced back to a diagnostic sample and 67% of these (4/6) were present as sub-clones.

Mutations in relapse-enriched genes comprised 33% (1/3) of the very early relapses, 50% (6/12) of the early and 40% (2/5) of the late relapses, which is quite comparable to the prevalence rates of 17%, 65% and 32% respectively, described by Li et al. [17]. Interestingly, in 3/4 (75%) of the *ETV6:RUNX1*-positive cases, the relapse was driven by a major clone in the epigenetic gene, *UHRF1*. Mutations in *UHRF1* make cells more sensitive to chemotherapy by reducing *FANC*-associated repair, but they also lead to global hypomethylation, pre-disposing patients to porto-oncogene activation, as was evident in the late *KRAS* mutation in one of our cases. However, the role of *UHRF1* as a surrogate marker of early relapse in *ETV6:RUNX1* cases needs to be prospectively studied.

The TA clonal pattern was predominant in 7/12 (58%) of the early relapses, with 71% (5/7) of these having major clonal mutations in the *NT5C2* or *PMS2* genes related to thiopurine-dose response. In addition, in 60% (3/5) of these cases, the *NT5C2/PMS2* clones were preceded by an initial hit in the epigenetic regulator, *KMT2D,* making it an important screening gene to predict early relapses during treatment. The *NT5C2* relapse-specific clonal mutations, R363L and R367Q, noted in two of our samples were already been shown by Li et al. to be functionally resistant to 6-mercaptopurine [17]. Loss-of-function mutations in the DNA mismatch repair gene, *PMS2*, have also been linked with thiopurine response in different studies [19,20,21]. All major mutant clones of *PMS2* noted at relapse were either nonsense or frameshift mutations, with one clone showing a high VAF, of 79%, later noted as having a loss of heterozygosity at Chr. 7p. However, none of the *PMS2*-driven relapse cases had evidence of underlying congenital mismatch repair-deficiency syndrome. Very early relapses were primarily m–M (2/3; 67%) originating from a minor *RAS* clone, which was consistent with previous evidence [17,31,32]. The late relapses had an equal frequency of M–M- and TA-type patterns, with M–M (2/4; 50%) classically associated with *Tp53* clones, which enter a quasi-dormant state under chemotherapy and result in late relapses (Figure 3a–e and Table 1). Overall, 14/46 (30%) of the samples showed a hypermutation phenotype based on criteria suggested by Waanders et al. in their study [27] (Table 1). The hypermutation frequency was 13% (6/46) on diagnosis and 17% (8/46) at relapse compared to the 3% at diagnosis observed by Waanders et al., although the relapse frequency was similar [26]. Further, the majority of the relapse cases with hypermutation developed from additional therapy-acquired clonal mutations (50%) or the persistence of a quiescent major clone at diagnosis (37.5%) (Table 1). Multiclonal mutations were noted in nine genes in fifteen cases (75%), with minor clonal *TENM3* or *NOTCH1* being very common (93%; 14/15) and showing persistence as minor clones at relapse. On the other hand, *KRAS*, *KMT2D*, *Flt3* and *FPGS* had both major and minor clones, either at diagnosis or relapse. Ma et al. noted multiclonal mutations in six genes in 50% (10/20) of their cases and only three of their genes (*KRAS*, *NRAS* and *PAX-5*) overlapped with our data [22].

The *CDKN2A/2B* deletions were the most commonly encountered CNVs, seen in 13/20 (65%) cases. However, only 46% of these (6/13) were either shared or present only at diagnosis, while 54% (7/13) were seen exclusively at relapse. Furthermore, *IKZF1* CNVs were seen in 25% of the cases (5/20) and 60% of these (3/5) were shared, which was similar to the findings of Kuiper et al. [30], who noted *IKZF1* CNVs in 23.5% of their cohort, of which 79% were shared.

To conclude, our study highlights a high frequency of early relapses driven by TA clones, suggesting the need for the development of an active surveillance strategy to identify their clonal rise. Ultra-deep sequencing (>3000x) in all the newly diagnosed B-ALL cases using a custom NGS panel for the *KRAS*, *NRAS*, *TP53*, *UHRF1* and *CREBBP* genes, followed by prospective testing of sequential samples (3-monthly) with a panel with additional *NT5C2*, *PMS2*, *KMT2D*, *ETV6* and *PAX5* genes will likely help to identify a significant proportion of cases with impending relapse, although this strategy might require dynamic modification based on prospective data emerging from the relapse biology in our setting.

## Figures and Tables

**Figure 1 diagnostics-13-00884-f001:**
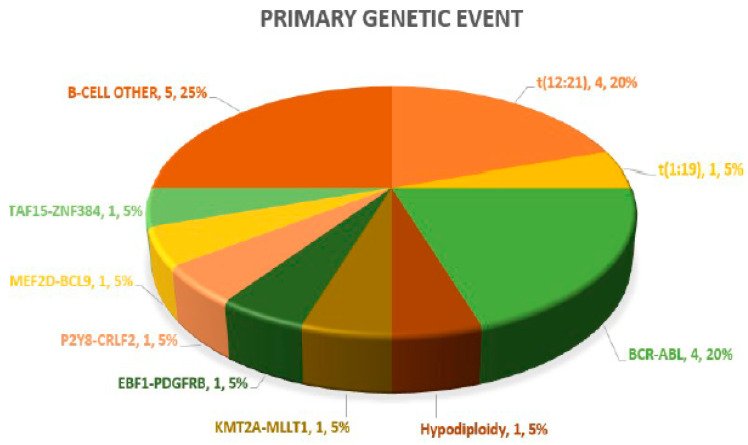
Pie chart highlighting spectrum of primary genetic abnormalities.

**Figure 2 diagnostics-13-00884-f002:**
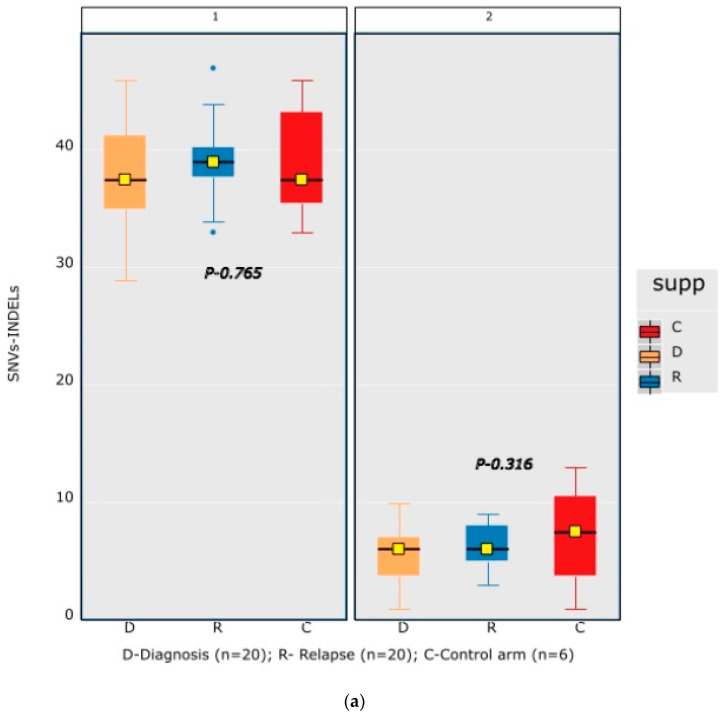
Comparative box and whisker plot for number of SNVs/INDELs in diagnosis (n = 20), relapse (n = 20) and non-relapse arm (without relapse n = 6) cases after first and second filter for Tier 1 and 2 somatic variants. Stacked bar chart that shows the number of major clones in different genes present at diagnosis only, shared or at relapse only. Histogram highlighting the different pathways and number of mutations noted in each at diagnosis or at relapse. Oncoprint map of combined SNVs/INDELs and CNVs noted in 20 paired-diagnosis–relapse B-ALL cases. Footnote for Figure 2d: ()-primary genetic event; []-total number of mutations including SNVs/INDELs and CNVs in each paired sample; {}-total number of mutations in the gene, along with breakdown as number of minor (m) or sub-clonal (<30%) and major (M) or clonal (>30%) mutations; red-colored genes denote the genes in which CNVs were noted. The *—represents the samples in which multiple minor or major mutations were noted in same gene. Details are as follows: C7—TENM3 (2 shared), NOTCH1 (2 relapse-specific); C17—TENM3 (1 shared, 1 diagnosis-specific), NOTCH1 (3 shared, 2 diagnosis-specific), EP300 (2 shared); C4— TENM3 (1 shared, 2 relapse-specific), NOTCH1 (1 shared, 1 diagnosis- and 1 relapse-specific); C8—TENM3 (2 shared); C13—KMT2D (1 shared, 2 diagnosis- and 1 relapse-specific); C18—TENM3 (4 relapse-specific), FPGS (1 shared and 1 relapse-specific); C9—TENM3 (3 shared, 1 relapse-specific), KMT2D (1 relapse- and 1 diagnosis-specific), PMS2 (1 diagnosis- and 1 relapse-specific); C10—TENM3 (2 relapse-specific), TET-2 (2 diagnosis-specific), KMT2D (2 relapse-specific), EP300 (1 diagnosis- and 2-relapse specific); C12- NOTCH1 (2 shared, 2 relapse-specific), KRAS (2 shared), KMT2D (2 shared); C14—TENM3 (2 shared), NOTCH1 (1 shared, 1 diagnosis-specific), FPGS (2 diagnosis-specific); C15- KMT2D (2 shared), TENM3 (2 diagnosis-specific); C19—NOTCH1 (3 diagnosis and 2 relapse-specific), TENM3 (3 relapse-specific), Flt3 (1 shared, 1 diagnosis-specific); C3- MSH6 (1 diagnosis- and 1 relapse-specific); C5—PAX5 (2 shared); C6 NOTCH1 (2 relapse-specific), TENM3 (2 relapse-specific); C11—TENM3 (3 diagnosis-specific).

**Figure 3 diagnostics-13-00884-f003:**
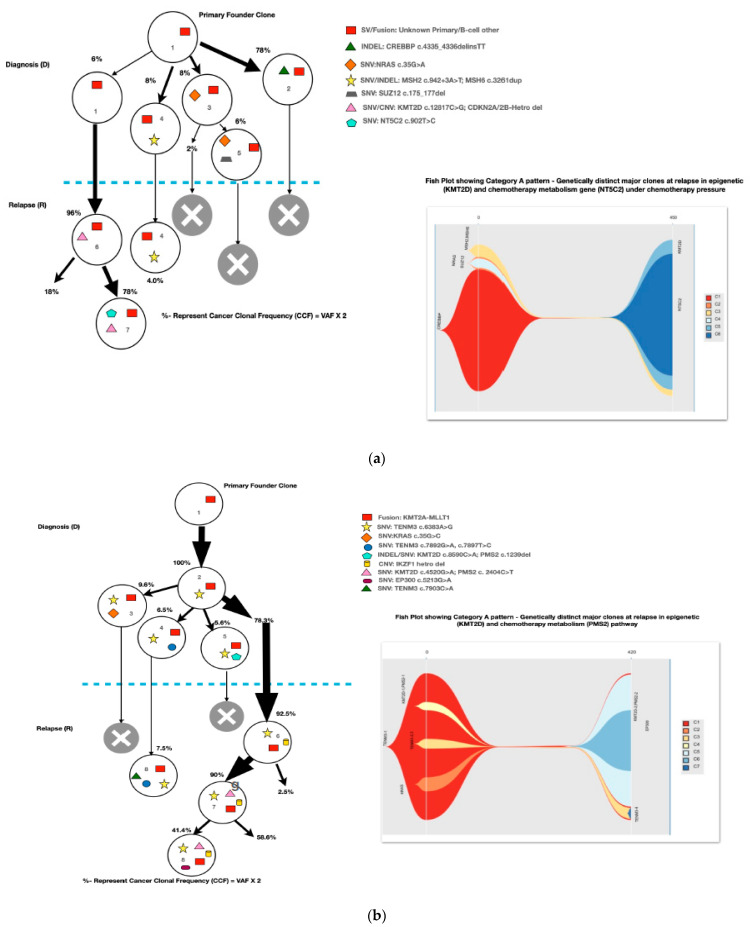
The clonal evolution patterns from diagnosis to relapse in different cases. Therapy-acquired clonal mutational pattern with *KMT2D* and *NT5C2* and *KMT2D* and *PMS2* clones at relapse, respectively. A minor clone of *NRAS* becoming a major clone at relapse. Genetically similar major clone to *Tp53* at diagnosis persisting at relapse. A mixed pattern of minor *KRAS* clone at diagnosis becoming a major clone at relapse, along with a therapy-acquired major clonal mutation in nucleotide metabolism (*NT5C2*) gene. The number % represent the clonal cellular fraction based on 2 x VAF %.

**Table 1 diagnostics-13-00884-t001:** Hematological, relapse, clonal evolution pattern and hypermutation-phenotype data in 20 paired B-ALL cases.

ID	Age/Gender	Primary Genetic Event	TLC (10^9/^L)	Day 35 MRD	Final Risk (ICiCLe)	Site of Relapse	Relapse Time (Months)	Relapse Type	Clonal Pattern	Pathways of Relapse (Major Gene Clone)	Hypermutation
C12	2 Y/M	*P2Y8-CRLF2*	56	Negative ≤ 0.01%	IR	BM	8	Very early	Minor–major	RAS pathway (*KRAS*)	No
C15	7 Y/M	*MEF2D-BCL9*	31.3	Negative ≤ 0.01%	SR	BM	8.5	Very early	Minor–major and therapy-acquired clone	RAS (*NRAS*) + nucleotide metabolism (*PRPS1*)	No
C18	7 Y/M	*BCR-ABL1*	51.2	Negative ≤ 0.01%	HR	BM + CNS	8	Very early	Unclassified	No major clones noted at diagnosis and relapse	No
C2	8 Y/M	Hypodiploidy	121	Positive 1.56%	HR	BM	12	Early	Therapy-acquired	Mismatch repair—thiopurine-dose response (*PMS2*)	No
C3	4 Y/F	B-cell other	182	Negative ≤ 0.01%	IR	CNS	15	Early	Therapy-acquired	Epigenetic (*KMT2D*) and nucleotide metabolism (*NT5C2*)	No
C4	5 Y/M	*BCR-ABL1*	96	Positive 0.76%	HR	BM + CNS	31	Early	Major–major	Epigenetic (*UHRF1*)	Yes
C5	3 Y/M	B-cell other	45.2	Negative ≤ 0.01%	SR	BM + CNS	31	Early	Major–major	B-cell development (*PAX-5*)	Yes
C6	4 Y/M	B-cell other	180	Negative ≤ 0.01%	IR	BM + CNS	31	Early	Minor–major and therapy acquired clone	RAS (*KRAS*) + nucleotide metabolism (*NT5C2*)	No
C8	8 Y/M	*BCR-ABL1*	29.6	Positive 3.5%	HR	BM	31	Early	Therapy-acquired	Mismatch repair—thiopurine-dose response (*PMS2*)	Yes
C9	10 Y/F	*KMT2A-MLLT1*	224	Negative ≤ 0.01%	HR	BM	14	Early	Therapy-acquired	Epigenetic (*KMT2D*) and mismatch repair—thiopurine-dose response (*PMS2*)	Yes
C10	5 Y/F	*EBF1-PDGFRB*	165	Positive 0.76%	HR	CNS	25	Early	Therapy-acquired	B-cell development (*ETV6*)	No
C13	10 Y/M	*BCR-ABL1*	67	Negative ≤ 0.01%	HR	BM + Testicular	30	Early	Therapy-acquired	Epigenetic (*KMT2D*) and nucleotide metabolism (*NT5C2*) and RAS pathways (*NRAS*)	Yes
C16	6 Y/M	*ETV6-RUNX1*	6.4	Positive 0.02%	HR	BM	20	Early	Major–major	Epigenetic (*UHRF1*) and RAS pathway (*KRAS*)	No
C17	3 Y/M	*ETV6-RUNX1*	8.1	Negative ≤ 0.01%	IR	BM	33	Early	Therapy-acquired	Epigenetic (*UHRF1*)	Yes
C19	11 Y/M	*TAF15-ZNF384*	20.5	Positive (0.1%)	HR	BM	32	Early	Minor–major	RAS Pathway (*Flt3*)	No
C1	5 Y/M	*ETV6-RUNX1*	41	Negative ≤ 0.01%	SR	BM	39	Late	Therapy-acquired	Epigenetic (*UHRF1*)	No
C7	6 Y/M	*ETV6-RUNX1*	5.5	Negative ≤ 0.01%	SR	BM	40	Late	Unclassified	-	Yes
C11	11 Y/M	B-cell other	30.1	Positive 0.02%	HR	BM	37	Late	Major–major	Epigenetic (*KMT2D*), RAS (*NRAS*) and Cell cycle (*Tp53*) clones increased in size at relapse	No
C14	4 Y/M	*TCF3-PBX1*	27	Negative ≤ 0.01%	SR	Testicular	38	Late	Therapy-acquired	RAS pathway (*KRAS*)	No
C20	5 y/M	B-cell other	150.6	Negative ≤ 0.01%	IR	BM + Testicular	37	Late	Major–major	Cell cycle (*Tp53*)	Yes

## Data Availability

Appendix A. In addition, raw data will be made available on request for non-commercial use.

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
