# Peer review of "Therapy-Acquired Clonal Mutations in Thiopurine Drug-Response Genes Drive Majority of Early Relapses in Pediatric B-Cell Precursor Acute Lymphoblastic Leukemia"

_diagnostics, 2023, doi:10.3390/diagnostics13050884_

Round 1

Reviewer 1 Report

Thakur et al. describe mutations in the leukemia cells in children with B-cell precursor acute lymphoblastic leukemias before treatment and in relpse. The presentation is alright overall, but has very little novelty value. Previous studies, e.g., publication cited as No 17, have presented similar data from many more patients and, at the same time, more in depth investigation of individual patients. In this respect, the data are of interest only if researchers seeking information for large pooled data sets in the future.

The data presented by the authors are confirmatory and do not allow for quantitative considerations on their own because the case numbers are too small. In this respect, they are also not a good indication of whether the data in the studied etnie are different from those in previously published collectives.

Notes in detail:

Table 1 is very confusing.

Table 2 is dispensable because the content is completely referenced in the text as well.

Figure 1: It is not understandable why in addition to the completely filtered data also the data after "filter 1" are presented.

Author Response

Response to Reviewer Comments

Reviewer 1 Comments

Q 1 : Thakur et al. describe mutations in the leukemia cells in children with B-cell precursor acute lymphoblastic leukemias before treatment and in relapse. The presentation is alright overall, but has very little novelty value. Previous studies, e.g., publication cited as No 17, have presented similar data from many more patients and, at the same time, more in depth investigation of individual patients. In this respect, the data are of interest only if researchers seeking information for large pooled data sets in the future.

The data presented by the authors are confirmatory and do not allow for quantitative considerations on their own because the case numbers are too small. In this respect, they are also not a good indication of whether the data in the studied etnie are different from those in previously published collectives.

Response 1:  We agree with reviewer that studies on this subject in larger cohorts are already published but we wish to add to that data considering relapse B-ALL being very heterogenous especially in regard to different ethnic regions and there being no study on this aspect of relapse B-ALL biology from our SEA Sub-continent. Moreover, the study does have novelty in that we used error read correction chemistry to minimise PCR bias and errors in data and very few studies have actually employed error red correction. Since, the results in a pilot cohort were interesting relating to high frequency of thiopurine dose modifying gene mutation sin early release we wish to record and publish our results. We plan to take the study forward with more numbers especially correlating with different primary genetic sub-types and as per sites of relapse. 

Q 2: Table 1 is very confusing.

Response 2: We have tried to re-organize data in table 1 to give it more clarity.

Q 3: Table 2 is dispensable because the content is completely referenced in the text as well.

Response 3: We agree and have removed table 2 to avoid duplication of data being presented.

Q 4: Figure 1: It is not understandable why in addition to the completely filtered data also the data after "filter 1" are presented.

Response 4: I think Figure 2a is being commented upon by reviewer as Figure 1 is a pie chart on frequency of different primary genetic event noted. Both filter 1 and filter 2 variant numbers were highlighted to show the change as well as any difference in mutation load pre and post filtering of data.

We thank the reviewer for their valuable time and effort in improving our manuscript.

Reviewer 2 Report

-first sentence of abstract should read DNA samples.

-line 34. CNS malignacies are leading  cause based on some reports.

-line 39-42. references are  missing. 

-table 1. patient with relapse on CNS or testis only are noted to have TA clonal evolution pattern. what samples were used at relapse?

-line 151. the ability NGS assay and used strategy to capture Ph-like ALL will require more elaboration (noted the reference but would to elaborate more).

-line 161. typo. 

-figure 2D. hard to read axis legends.

-line 206 enriched.

-Definition, criteria and validation for TA clones should be included and discussed more in the methods / introduction.

-Therapy-induced mutations drive the genomic landscape of relapsed acute lymphoblastic leukemia by Bashing Li should referenced and contrasted  to current approach.  Also Upfront Treatment Influences the Composition of Genetic Alterations in Relapsed Pediatric B-Cell Precursor Acute Lymphoblastic Leukemia by Roland Kuiper should be referenced and contrasted with current study. 

-Are the authors suggesting that the screening in blood every 3 months with their assay can detect relapsed clones? what would be the limit of sensitivity and how that would compare with MRD flow cytometry?

-reference format is inconsistent.

Author Response

Response to Reviewer Comments 

Reviewer 2

Q 1: First sentence of abstract should read DNA samples.

Response 1: Edited as per reviewer observation.

Q 2: Line 34. CNS malignancies are leading  cause based on some reports.

Response 2: We agree and have re-edited the sentence to “ one of the “  foremost causes.

Q 3: line 39-42. references are  missing. 

Response 3: We have provided the references at end of line 43 as it is in continuation to earlier lines. 

Q 4: Table 1. patient with relapse on CNS or testis only are noted to have TA clonal evolution pattern. what samples were used at relapse?

Response 4: We used testicular FNAC and or CSF centrifuged pellet for DNA extraction and analysis. 

Q 5: Line 151. the ability NGS assay and used strategy to capture Ph-like ALL will require more elaboration (noted the reference but would to elaborate more).

Response 5: Elaborated in methods as suggested.

Q 6: line 161. typo. 

Response 6: Corrected.

Q 7: figure 2D. hard to read axis legends.

Response 7: The figure with better resolution is being added.

Q 8: line 206 enriched.

Response 8: Corrected.

Q 9: Definition, criteria and validation for TA clones should be included and discussed more in the methods / introduction.

Response 9: The same has been detailed in Introduction and methodology in relevant sections. 

Q 10: Therapy-induced mutations drive the genomic landscape of relapsed acute lymphoblastic leukemia by Bashing Li should referenced and contrasted  to current approach.  Also Upfront Treatment Influences the Composition of Genetic Alterations in Relapsed Pediatric B-Cell Precursor Acute Lymphoblastic Leukemia by Roland Kuiper should be referenced and contrasted with current study. 

Response 10: The reference for Li et al was already discussed and mentioned in the manuscript as reference 17. We have added the study reference for Kuiper et al at reference 30 and adequately compared our findings with their important findings in discussion at relevant sections. 

Q 11: Are the authors suggesting that the screening in blood every 3 months with their assay can detect relapsed clones? what would be the limit of sensitivity and how that would compare with MRD flow cytometry?

Response 11: We believe prospective validation of this approach can help us pick up clones especially in cases that are likely to have early relapses. Digital PCR based probes can help detect a clone upto 10-5 to 10-6 level, which is log 10 better than current flow cytometry based MRD approach. This has been demonstrated in few western studies and needs to be seen how it can be helpful or replicated in our resource constraint setting. 

Q 12: reference format is inconsistent.

Response 12: We have re-checked and edited the references as per consistent journal format. 

We thank the reviewer for their constructive comments to improve our manuscript quality.